# Population Pharmacodynamic Models of Risperidone on PANSS Total Scores and Prolactin Levels in Schizophrenia

**DOI:** 10.3390/ph17020148

**Published:** 2024-01-23

**Authors:** Zhiwei Huang, Lei Zhang, Yan Li, Yimin Yu, Yifeng Shen, Xiujia Sun, Kun Lou, Hongmei Luo, Zhibin Meng, Huafang Li, Yumei Wei

**Affiliations:** 1Shanghai Mental Health Center, Shanghai Jiao Tong University School of Medicine, Shanghai 200030, China; hzw_mail@163.com (Z.H.);; 2CSPC Zhongqi Pharmaceutical Technology (Shijiazhuang) Co., Ltd., Shijiazhuang 050000, China; 3Shanghai Clinical Research Center for Mental Health, Shanghai 200030, China; 4Shanghai Key Laboratory of Psychotic Disorders, Shanghai 200030, China

**Keywords:** population pharmacodynamics, PANSS total scores, prolactin, risperidone, schizophrenia

## Abstract

Currently, research predominantly focuses on evaluating clinical effects at specific time points while neglecting underlying patterns within the treatment process. This study aims to analyze the dynamic alterations in PANSS total scores and prolactin levels in patients with schizophrenia treated with risperidone, along with the influencing covariates. Using data from an 8-week randomized, double-blind, multicenter clinical trial, a population pharmacodynamic model was established for the PANSS total scores of and prolactin levels in patients treated with risperidone. The base model employed was the E_max_ model. Covariate selection was conducted using a stepwise forward inclusion and backward elimination approach. A total of 144 patients were included in this analysis, with 807 PANSS total scores and 531 prolactin concentration values. The PANSS total scores of the patients treated with risperidone decreased over time, fitting a proportionally parameterized sigmoid E_max_ model with covariates including baseline score, course of the disease, gender, plasma calcium ions, and lactate dehydrogenase levels. The increase in prolactin levels conformed to the ordinary E_max_ model, with covariates encompassing course of the disease, gender, weight, red blood cell count, and triglyceride levels. The impacts of the baseline scores and the course of the disease on the reduction of the PANSS scores, as well as the influence of gender on the elevation of prolactin levels, each exceeded 20%. This study provides valuable quantitative data regarding PANSS total scores and prolactin levels among patients undergoing risperidone treatment across various physiological conditions.

## 1. Introduction

Schizophrenia is widely regarded as one of the most severe mental disorders among all psychiatric illnesses, often necessitating prolonged treatment with antipsychotic medication [1]. It manifests as a confluence of positive symptoms, including hallucinations and delusions; negative symptoms, like avolition, alogia, and expressive deficits; and cognitive impairments, such as memory deficits and executive dysfunction [2]. The *Diagnostic and Statistical Manual of Mental Disorders*, fourth edition, text revision (DSM-IV-TR), classifies schizophrenia as paranoid, catatonic, simple, undifferentiated, disorganized, and residual [3]. The most prevalent subtype is paranoid schizophrenia, primarily distinguished by the presence of delusions and heightened suspicion. Atypical antipsychotic medications have become the most commonly used drugs for treating schizophrenia [4]. Among these medications, risperidone is a widely utilized medication, primarily exerting its antipsychotic effects through the blocking of the dopamine D2 and serotonin 5-HT2A receptors, though it carries a higher incidence of extrapyramidal symptoms and hyperprolactinemia adverse reactions [5,6,7].

The severity of schizophrenia in patients can be measured using the Positive and Negative Syndrome Scale (PANSS). The PANSS scoring comprises positive subscale scores (P1–P7), negative subscale scores (N1–N7), and general psychopathology scores (G1–G16) [8]. The total score range is 30–210 points, with higher scores indicating more severe conditions. Meta-analyses have shown that PANSS scores tend to decrease more significantly in the treatment summaries of female patients, patients with more severe symptoms at baseline, patients not on antipsychotic medications, and patients with shorter courses of the disease [9]. Prolactin is a polypeptide hormone secreted by the anterior pituitary cells. Risperidone’s potent blocking of the D2 receptors may lead to hypothyroidism and increased secretion of prolactin from the pituitary [10,11]. Elevated prolactin levels could trigger various issues such as menstrual irregularities in females, osteoporosis, gynecomastia in males, and sexual dysfunction and significantly impact the overall quality of life of patients. It may also result in adjustments to the dosages and types of medications prescribed to patients [12]. The condition characterized by elevated levels of prolactin beyond the established normal range is referred to as hyperprolactinemia. It is commonly accepted that the normal range for women is 15–25 µg/L, while for men, it is 15–20 µg/L. Patients undergoing antipsychotic drug treatment exhibit high susceptibility to hyperprolactinemia, with an incidence rate ranging from 70% to 90%, particularly in the case of risperidone and its metabolite, paliperidone (9-OH risperidone) [13]. The elevation of prolactin levels is influenced not only by factors such as age and gender but also by dosages of risperidone and the duration of treatment [14]. The existing intervention approaches exhibit significant limitations, including the utilization of supplementary antipsychotic medications and symptomatic hormone therapy, both of which possess inherent drawbacks. Furthermore, these intervention strategies do not consistently result in successful outcomes and may even induce a recurrence of mental symptoms, thereby contributing to unstable conditions.

The duration of risperidone treatment is associated with the relief of symptoms and the occurrence of adverse reactions in patients with schizophrenia. Currently, research is primarily concentrating on assessing specific time points, disregarding the underlying patterns of disease progression. However, employing population pharmacodynamic (PD) modeling can provide a more comprehensive understanding of the dynamics of PANSS scores and prolactin levels over time. The E_max_ model is a common tool to make quantitative evaluations for time–effect relationships, which can help achieve accurate predictions of pharmacodynamics from different patients [15]. Based on data from a multicenter, randomized, double-blind study, this research establishes a population pharmacodynamic model for PANSS total scores and prolactin levels in patients treated with risperidone, elucidating the temporal trajectory of changes and the influencing covariates.

## 2. Results 

### 2.1. Demographic Data 

A total of 144 patients receiving risperidone were included in this study, with 123 patients completing the 8-week follow-up, as shown in Figure 1. Most of the patients were young, with a median age of 33, and approximately half of them were female. Out of 73 female patients, 10 were post-menopausal. The median course of the disease was 5 years. The patients had relatively severe baseline conditions, with a median PANSS score of 90. The baseline prolactin level was relatively low, with a median value of 21.6 ng/mL. The baseline demographic characteristics of the subjects are shown in Table 1.

### 2.2. Population Pharmacodynamic Analysis of PANSS Total Scores

A parameterized sigmoid E_max_ model was established based on 807 PANSS scores from the 144 schizophrenia patients, adequately describing the change in the scores over time. The mean reduction rate of the PANSS was 64.9 ± 22.1%. Specifically, there was a decrease of 13.6 ± 5.7 in the PANSS positive scores, a decrease of 7.9 ± 4.9 in the PANSS negative scores, and a decrease of 17.4 ± 7.7 in the PANSS general psychopathology scores. The parameterized E_max_ model represented a proportional decrease in the maximum effect with baseline E_0_. The correlation coefficient between E_0_ and the Hill coefficient (γ) was 0.389. This suggests that patients with higher baseline scores not only experienced numerically greater reductions in PANSS total scores but also exhibited steeper declines in these scores over time.

In the covariate screening analysis, plasma calcium had a significant impact on γ, while the course of the disease and gender significantly affected the time to 50% E_max_ (ET_50_) and lactate dehydrogenase (LDH) significantly affected E_max_, as shown in Table 2. The final model can be described as follows: patients with higher blood calcium levels had larger γ values, indicating a steeper decline in their PANSS total scores. Patients with shorter disease durations and female patients had shorter ET_50_ values, indicating a quicker onset of the therapeutic effect. The patients with higher LDH levels had larger proportions of reduction in the PANSS total scores. The impact of these covariates on the PD model parameters and PANSS scores is illustrated in Figure 2. The impact of the baseline scores and the course of the disease on the reduction of the PANSS scores exceeded 20%.
(1)$$E_{0}=90.1\times e^{\eta_{1}}$$
(2)$$\gamma =1.31\times {\left(Ca/2.3\right)}^{2.56}\times e^{\eta_{2}}$$
(3)$${ET}_{50}=5.37\times {\left(CD/5\right)}^{0.267}\times 0.726^{Female}\times e^{\eta_{3}}$$
(4)$$E_{max}=0.661\times {\left(LDH/155\right)}^{0.191}$$

Ca: plasma calcium; CD: course of the disease; E_0_: baseline; E_max_: maximum effect; ET_50_: time to 50% E_max_; LDH: lactate dehydrogenase; γ: Hill coefficient; and η: inter-individual variability.

### 2.3. Population Pharmacodynamic Analysis of Prolactin

An ordinary E_max_ model was established using 531 prolactin concentration values from 144 patients receiving risperidone. ET_50_ was fixed at the empirical value of 0.10, and this model adequately described changes in prolactin levels over time. In the covariate screening analysis, the course of the disease, the red blood cell count, triglycerides, and gender significantly affected E_0_, while weight and gender significantly affected E_max_, as shown in Table 3. The final model can be described as follows: patients with shorter disease durations, lower red blood cell counts, and higher triglycerides, as well as females, had larger E_0_ values, indicating higher baseline prolactin levels. Female patients with higher weights had larger absolute increases in prolactin levels. The impacts of these covariates on the PD model parameters and prolactin levels are illustrated in Figure 3. The impact of gender on the increase in prolactin levels exceeded 20%.
(5)$$E_{0}=15.0\times {\left(CD/5\right)}^{-0.162}\times {\left(RBC/4.5\right)}^{-2.71}\times {\left(TG/0.9\right)}^{0.392}\times 1.62^{Female}\times e^{\eta_{1}}$$
(6)$${ET}_{50}=0.10$$
(7)$$E_{max}=34.0\times {\left(WT/60\right)}^{-0.570}\times 3.11^{Female}\times e^{\eta_{2}}$$

CD: course of the disease; E_0_: baseline; E_max_: maximum effect; ET_50_: time to 50% E_max_; RBC: red blood cell; TG: triglyceride; WT: weight; and η: inter-individual variability.

### 2.4. Model Evaluation

The goodness of fit (GOF) plots, as shown in Appendix A, demonstrate good consistency, with the observed values evenly distributed on both sides of the *Y* = *X* axis for both population prediction (PRED) and individual prediction (IPRED) data. The distribution of conditional weighted residuals (CWRESs) appears to be symmetrical, with most values falling within the range of −4 and +4. The visual predictive check (VPC), as depicted in Figure 4 and Figure 5, overall shows that the 2.5, 50, and 97.5 percentiles of the observed values are contained within the 95% prediction intervals of these percentiles, indicating accurate predictions of PANSS scores and prolactin levels by the model at different visit times. The success rates of the bootstrap were 99.0% and 95.6%, respectively. The estimated parameters of the final model closely align with the medians obtained through bootstrapping and are all within the 95% confidence intervals (CIs), as presented in Table 2 and Table 3.

### 2.5. Predictions

We simulated the decreasing trend of PANSS total scores over an 8-week period for the typical patient, with baseline scores of 79 or 104 and a disease course of 0.5 years or 20 years, as shown in Figure 2C. The prolactin levels during those 8 weeks for typical male and female patients are shown in Figure 3C.

## 3. Discussion

The E_max_ model aptly delineated the temporal curves of both PANSS scores and prolactin levels in schizophrenia patients undergoing risperidone treatment, with model diagnostics and VPCs showcasing the robustness and predictive performance of the model.

During the modeling process of the PANSS scores, a strong correlation was observed between parameters E_max_ and E_0_, with a correlation coefficient of 1.00, leading to a failure in the covariance matrix computation. Therefore, a reparametrization of the E_max_ model was performed by aligning E_max_ proportionally with E_0_, which enhanced the model’s robustness [16]. The first post-administration prolactin concentration sampling point was set at the end of week 2, by which time the patients’ prolactin levels had swiftly reached a plateau. The absence of an intervening rise made it challenging to estimate ET_50_ accurately [17]. Consequently, during the modeling of the prolactin levels, ET_50_ was fixed at a minor empirical value of 0.10 [18]. Simplifying the sigmoid E_max_ model to an ordinary E_max_ model reduced the OFV by 4.32, suggesting a better fit with the ordinary E_max_ model.

Several studies have identified that patients with shorter courses of the disease and female patients tend to respond more rapidly to treatment [19,20], aligning with the findings of this study. One hypothesis posits that schizophrenia may be triggered by reduced Ca^2+^ entry through the N-methyl-D-aspartate (NMDA) receptors [21]. Numerous studies have highlighted the antimanic and mood-stabilizing properties of calcium channel blockers, particularly verapamil and nimodipine [22,23]. Isradipine demonstrated efficacy in ameliorating the symptoms of schizophrenia in two recent randomized controlled trials [24,25]. This study discovered that higher plasma Ca^2+^ concentrations steepened the decline curve of PANSS scores, likely chiefly contributed to by a reduction in positive symptom scores. Hence, it is speculated that the co-administration of calcium channel blockers might yield an adjuvant impact for patients with evident positive symptoms. However, the existing clinical data remain inadequate or primarily anecdotal, with a scarcity of systematic evidence. Additional clinical investigations are imperative to delve into this matter. 

It is known that LDH serves as a biomarker for cellular death [26]. Schizophrenia elevates cellular death levels, leading to an increase in circulating cell-free DNA (cfDNA) concentrations [27]. This study found that patients with higher LDH levels exhibited stronger therapeutic effects from risperidone treatment, which might be linked to the cellular death pathway. The presently available data and literature do not provide sufficient evidence to establish a definitive causal association between these two factors. On the other hand, LDH functions as an enzyme in the glycolytic energy pathway, facilitating the conversion of pyruvate to lactate and serving as a pivotal point where crucial energy metabolism pathways converge. It has been reported that diminished concentrations of LDH in the cerebrospinal fluid may serve as potential biomarkers for prodromal and negative symptoms (especially social withdrawal) [28]. A decline in LDH expression in the cerebrospinal fluid signifies aberrant or potentially impaired cerebral energy generation. At present, there exists a lack of correlation data pertaining to concentrations of cerebrospinal fluid LDH and plasma LDH.

It is well-known that baseline prolactin levels and prolactin elevations vary vastly between males and females due to intrinsic physiological differences. Prolactin elevation can also induce weight gain [29]. Through synergism with erythropoietin, prolactin can stimulate red blood cell production [30]. In another animal study, it was reported that prolactin increased the number of mature red blood cells at the expense of white blood cell production [31]. Antipsychotic treatment often coincides with simultaneous elevations in prolactin and triglycerides [32]. An observational study revealed a positive correlation between prolactin levels and triglycerides in breast cancer patients [33]. Conversely, two other studies demonstrated a negative correlation between prolactin levels and triglycerides in patients diagnosed with polycystic ovary syndrome and those who were overweight or obese [34,35]. This study identified red blood cell count, triglycerides, and weight as covariates in the prolactin model, but further investigation is needed to elucidate the causal relationships.

This study has several limitations. The short follow-up duration of 8 weeks restricted generalization to patients undergoing long-term treatment. The PANSS score model did not differentiate between positive and negative symptoms. Scatter plots and the LOESS lines of the PANSS positive scores, PANSS negative scores, PANSS general psychopathology scores, and PANSS total scores are shown in Appendix A. The significant decrease in the PANSS positive scores might be attributed to the selection criteria, which required participants to have at least two out of seven items on the PANSS positive scores ≥ 4 points. The initial post-administration prolactin concentration sampling point, set at week 2, hindered the accurate acquisition of the prolactin’s rising ET_50_. Our present study did not measure the patients’ plasma risperidone concentrations, precluding analysis of the relationship between drug exposure and PANSS scores or prolactin levels.

## 4. Materials and Methods 

### 4.1. Study Population

This study was randomized, double-blind, dual-simulated, parallel-controlled, multicenter research investigating the treatment of schizophrenia with risperidone and iloperidone tablets, randomized with DAS (version 3.0). It received approval from the Ethical Committee of Shanghai Mental Health Center (approval No. 2013-42). This study adhered to the Good Clinical Practice (GCP) guidelines and the Declaration of Helsinki, with informed consent co-signed by all participants and their legal guardians. It included inpatient participants aged 18 to 65 years, meeting the DSM-IV clinical diagnostic criteria for schizophrenia, with no gender restrictions. Additionally, the patients were required to have PANSS total scores of ≥70 points, with at least two out of seven items on the PANSS Positive Symptom Subscale (PANSS-P) scoring ≥4 points. The primary exclusion criteria were as follows: patients who had previously been treated with full doses and full courses of at least two different antipsychotic drugs without satisfactory improvement; those with poor efficacy from prior full-dose full-course treatment with risperidone; patients with histories of organic brain diseases such as Parkinson’s disease, histories of tardive dyskinesia, or neuroleptic malignant syndrome or histories of epilepsy; patients with severe excitation, tendencies to harm others or suicidal tendencies; patients who had systematically used chlorpromazine within 3 months before the baseline; patients who had used antipsychotic or antidepressant medications before the baseline without a washout period (at least 5 half-lives and at least 1 day); patients who had received electroconvulsive therapy (ECT) and systemic psychotherapy within 60 days before screening; pregnant or lactating women; and patients with clinically significant abnormal laboratory findings, as judged by the study physician.

### 4.2. Clinical Data

This analysis includes only the data of the patients in the risperidone administration group. Risperidone tablets of 1 mg/tablet, batch number 130607932, were produced by Xi’an Janssen Pharmaceutical Ltd. Dosage adjustments were made between 3 and 6 mg/day based on the therapeutic efficacy and tolerance of the subjects over a treatment period of 8 weeks. The PANSS was evaluated once at the screening stage and the end of weeks 1, 2, 4, 6, and 8 post-administration, while prolactin was checked once at the screening stage and at the end of weeks 2, 4, and 8 post-administration.

### 4.3. Modeling Approach

Models were developed using the first-order conditional estimation method with interaction in NONMEM 7 (version 7.5.0), a nonlinear mixed effects model software. Perl-speaks-NONMEM (PsN), version 5.0.0, and MaS Studio, version 1.6.0.5, were utilized as auxiliary software.

The reduction in PANSS scores was described using a proportionally parameterized sigmoid E_max_ model, as shown in Equation (8), while the increase in prolactin was described using an ordinary E_max_ model, as shown in Equation (9).
(8)$$E=E_{0}\cdot (1-\frac{E_{max}\cdot {Time}^{\gamma }}{{ET}_{50}^{\gamma }+{Time}^{\gamma }})$$
(9)$$E=E_{0}+\frac{E_{max}\cdot Time}{{ET}_{50}+Time}$$

It was assumed that the inter-individual variability of the PD model parameters followed a log-normal distribution, as shown in Equation (10):(10)$$\theta_{i}=\theta_{T}\cdot e^{\eta_{i}}$$
where *θ_i_* represents the PD model parameter for the *i*-th subject; *θ_T_* is the typical population value of the PD model parameter; and *η_i_* is normally distributed, with a mean of 0 and a variance of *ω*^2^. Residual variability was addressed using additive or mixed residual error models, as shown in Equations (11) and (12):(11)$$Y_{ij}={IPRED}_{ij}+\varepsilon_{A,ij}$$
(12)$$Y_{ij}={IPRED}_{ij}\cdot (1+\varepsilon_{P,ij})+\varepsilon_{A,ij}$$
where *Y_ij_* and *IPRED_ij_* represent the observed and predicted values, respectively, for the *i*-th subject at the *j*-th time point; *ε*_*p*,*ij*_ and *ε*_*A*,*ij*_ represent the proportional and additive residuals, respectively, for the *i*-th subject at the *j*-th time point, with both being independent and normally distributed with means of 0 and variances of *σ*_1_^2^ and *σ*_2_^2^, respectively.

Potential covariates were screened through stepwise forward inclusion and backward elimination at significance levels of *p* < 0.05 and *p* < 0.01, respectively: i.e., a decrease in the objective function value (OFV) by more than 3.84 and an increase in the OFV by more than 6.63 (*χ*^2^ distribution with 1 degree of freedom). The covariates under consideration included gender, age, weight (WT), course of the disease (CD), age at onset, whether the subject was a first-episode patient, concomitant use of sleep medications, concomitant use of other antipsychotic medications, risperidone dosage, white blood cell count, red blood cell (RBC) count, hemoglobin content, platelet count, neutrophil count, cholesterol, triglycerides (TGs), alanine aminotransferase, aspartate aminotransferase, creatinine, creatine phosphokinase, blood urea nitrogen, LDH, blood glucose, direct bilirubin, total bilirubin, calcium (Ca) ions, potassium ions, phosphate ions, chloride ions, and sodium ions.

The impact of the continuous covariates on the PD model parameters was illustrated as is shown in Equation (13), while the impact of the categorical covariates on those parameters is shown in Equation (14):(13)$$\theta_{i}=\theta_{T}\cdot {\left({Cov}_{i}/{Cov}_{median}\right)}^{\theta_{cov}}$$
(14)$$\theta_{i}=\theta_{T}\cdot {\theta_{cov}}^{X_{i}}$$
where *θ_i_* is the model parameter value for the *i*-th subject; *θ_T_* is the typical value of the model parameter; *Cov_i_* is the value of the continuous covariate for the *i*-th subject; *Cov_median_* is the median value of the continuous covariate in the population; *X_i_* is the category of the categorical covariate for the *i*-th subject, where 0 represents the more prevalent category and 1 represents the other category; and *θ_cov_* is the coefficient describing the size of the covariate effect.

### 4.4. Model Evaluation

The model evaluation included GOF, VPCs, and bootstrap. The GOF included PRED vs. the dependent variable (DV), IPRED vs. the DV, CWRES vs. PRED, and CWRES vs. time. VPC simulation, 1000 times, and bootstrap extraction, 1000 times, were used to verify the accuracy and robustness of the models.

### 4.5. Predictions

The E_max_ model formula was utilized to simulate the PANSS total score and prolactin level curves of the final model under the influences of various covariates. The 10th and 90th percentiles of the continuous covariates were used for simulation.

## 5. Conclusions

In summary, this study provides valuable quantitative data regarding PANSS total scores and prolactin levels among patients undergoing risperidone treatment across various physiological conditions. Baseline PANSS total scores and the course of the disease have over a 20% impact on the reduction of PANSS scores, while gender has over a 20% impact on the elevation of prolactin levels. It is advisable to consider personalized durations of risperidone treatment for patients with varying baseline PANSS total scores and disease courses.

## Figures and Tables

**Figure 1 pharmaceuticals-17-00148-f001:**
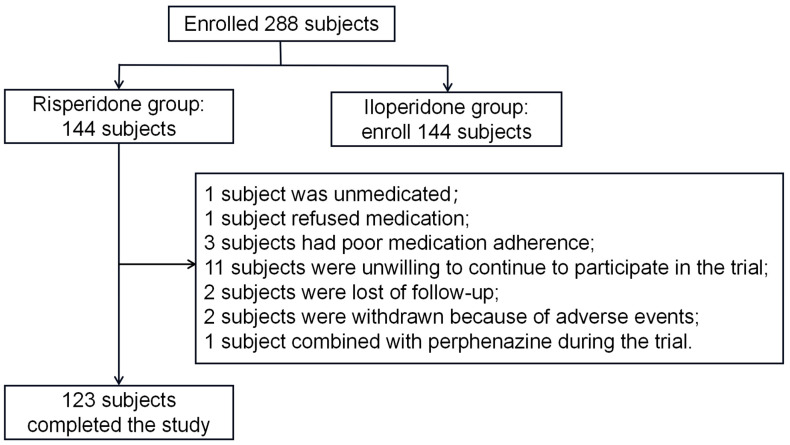
Flowchart of the enrolled patients.

**Figure 2 pharmaceuticals-17-00148-f002:**
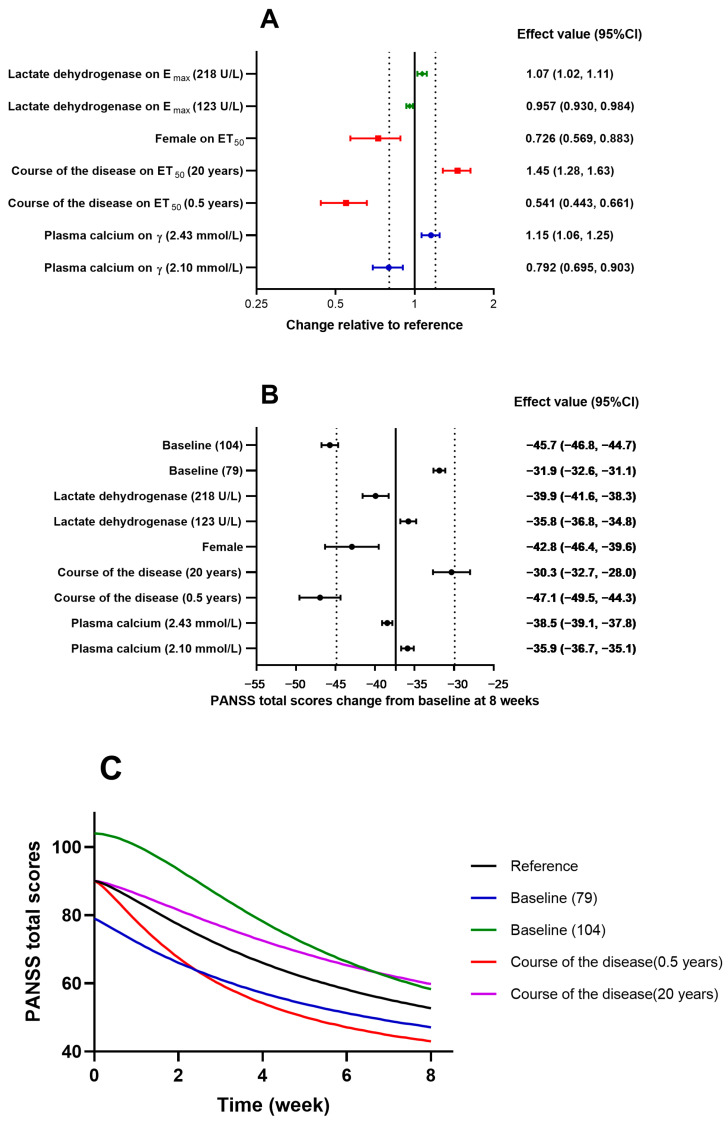
Covariate effect evaluation of the population pharmacodynamic model of PANSS total scores. (**A**) Effects on pharmacodynamic parameters are expressed relative to a reference patient (male, with baseline plasma calcium = 2.3 µmol/L, course of the disease = 5 years, and lactate dehydrogenase = 155 U/L). (**B**) Effects on PANSS total score changes at week 8 are expressed relative to the reference patient. (**C**) Simulated PANSS total score changes of a typical patient over treatment time. Data are estimates (95% CI). Dotted lines represent the reference ±20%. CI: confidence interval; E_0_: baseline; E_max_: maximum effect; ET_50_: time to 50% E_max_; PANSS: Positive and Negative Syndrome Scale; γ: Hill coefficient.

**Figure 3 pharmaceuticals-17-00148-f003:**
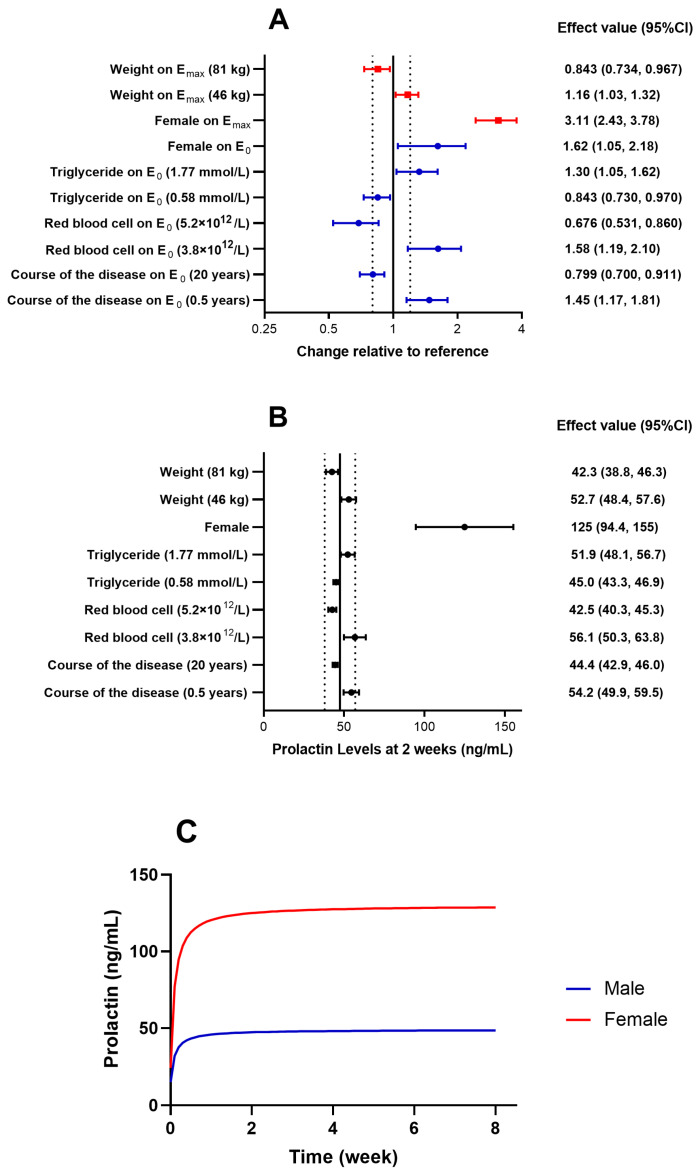
Covariate effect evaluation of the population pharmacodynamic model of prolactin levels. (**A**) Effects on pharmacodynamic parameters are expressed relative to a reference patient (male, with baseline course of the disease = 5 years, red blood cells = 4.5 × 10^12^/L, triglycerides = 0.9 mmol/L, and weight = 60 kg). (**B**) Effects on prolactin levels change at week 2 are expressed relative to the reference patient. (**C**) Simulated prolactin level changes of a typical patient over treatment time. Data are estimates (95% CI). Dotted lines represent the reference ±20%. CI: confidence interval; E_0_: baseline; E_max_: maximum effect.

**Figure 4 pharmaceuticals-17-00148-f004:**
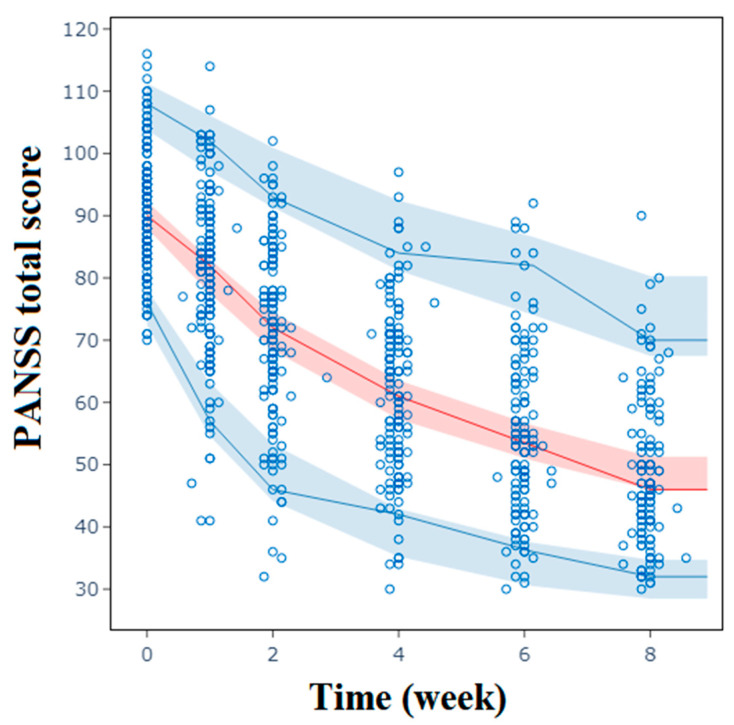
Visual predictive check for the final population pharmacodynamic model of PANSS total scores. Blue circles are the observations. Red solid lines represent the observed median; blue solid lines represent the observed 2.5th and 97.5th percentiles. Shaded bands are simulation-based 95% prediction intervals for the median, 2.5th, and 97.5th percentiles. PANSS: Positive and Negative Syndrome Scale.

**Figure 5 pharmaceuticals-17-00148-f005:**
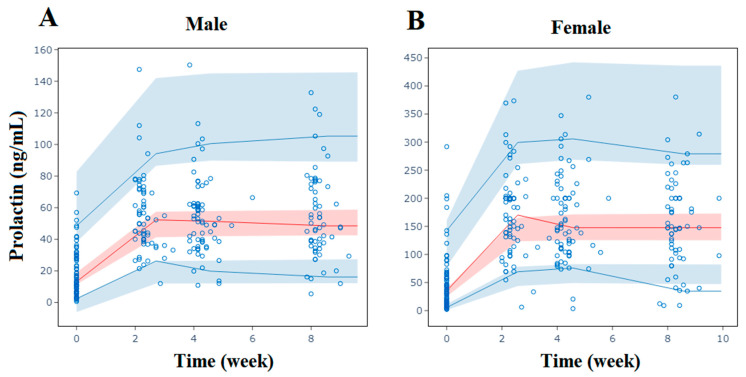
Visual predictive check for the final population pharmacodynamic model of prolactin levels. Blue circles are the observations. Red solid lines represent the observed median; blue solid lines represent the observed 2.5th and 97.5th percentiles. Shaded bands are simulation-based 95% prediction intervals for the median, 2.5th, and 97.5th percentiles.

**Table 1 pharmaceuticals-17-00148-t001:** Demographic and clinical characteristics in the risperidone therapy group (*n* = 144).

	Median (Interquartile Range)	Min–Max
Male (%)	49.3	
Age (years)	33.3 (25.2–45.2)	18.3–63.4
Height (cm)	165 (160–172)	150–183
Weight (kg)	59.3 (52.0–70.0)	37.0–106.0
Baseline Prolactin (ng/mL)	21.6 (10.2–42.6)	0.596–292
Course of Disease (Years)	5.00 (1.00–8.00)	0.500–38.0
Baseline PANSS Total Scores	90.0 (84.0–97.0)	70.0–116.0

**Table 2 pharmaceuticals-17-00148-t002:** Parameter estimates for the final population pharmacodynamic model of PANSS total scores.

Parameter	NONMEM		Bootstrap	
	Estimate	95% CI	Median	95% CI
Fixed Effect				
E_0_	90.1	88.5, 91.7	90.1	88.5, 91.7
γ	1.31	1.18, 1.44	1.31	1.17, 1.45
ET_50_	5.37	4.43, 6.31	5.42	4.46, 6.60
E_max_	0.661	0.631, 0.691	0.663	0.630, 0.711
Covariate Effect				
Plasma Calcium on γ	2.56	1.12, 4.00	2.56	0.880, 4.01
Course of Disease on ET_50_	0.267	0.180, 0.354	0.262	0.169, 0.355
Sex on ET_50_	0.726	0.569, 0.883	0.725	0.571, 0.913
Lactate Dehydrogenase on E_max_	0.191	0.0689, 0.313	0.197	0.0410, 0.347
Inter-Individual Variability				
E_0_ (%)	10.1	8.91, 11.3	10.1	8.84, 11.3
γ (%)	40.7	31.9, 49.5	40.3	31.5, 50.0
ET_50_ (%)	62.2	51.4, 73.0	60.9	50.6, 73.1
ρ(E_0_, γ)	0.389	0.154, 0.624	0.396	0.205, 0.455
Residual Error				
Additive Error	3.78	3.43, 4.13	3.77	3.41, 4.11

E_0_: baseline; E_max_: maximum effect; ET_50_: time to 50% E_max_; γ: Hill coefficient; ρ: correlation coefficient.

**Table 3 pharmaceuticals-17-00148-t003:** Parameter estimates for the final population pharmacodynamic model of prolactin levels.

Parameter	NONMEM		Bootstrap	
	Estimate	95% CI	Median	95% CI
Fixed Effect				
E_0_	15.0	11.6, 18.3	14.9	11.9, 18.5
ET_50_	0.100 FIX			
E_max_	34.0	28.8, 39.2	34.2	28.9, 39.4
Covariate Effect				
Course of Disease on E_0_	−0.162	−0.257, −0.0675	−0.163	−0.266, −0.0592
Red Blood Cells on E_0_	−2.71	−4.38, −1.04	−2.75	−4.29, −1.09
Triglycerides on E_0_	0.392	0.0685, 0.716	0.405	0.0819, 0.750
Sex on E_0_	1.62	1.05, 2.18	1.63	1.15, 2.26
Sex on E_max_	3.11	2.43, 3.78	3.09	2.48, 3.87
Weight on E_max_	−0.570	−1.03, −0.112	−0.566	−1.10, −0.105
Inter-Individual Variability				
E_0_ (%)	70.0	56.3, 81.4	68.4	56.2, 82.9
E_max_ (%)	49.7	38.4, 58.9	48.8	38.7, 59.8
Residual Error				
Proportional Error (%)	25.0	18.5, 29.4	25.0	19.5, 29.1
Additive Error (ng/mL)	6.70	2.79, 9.06	6.67	3.35, 9.05

E_0_: baseline; E_max_: maximum effect; ET_50_: time to 50% E_max._

## Data Availability

The raw data supporting the conclusion of this article will be made available by the authors, without undue reservation.

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
