# Peer review of "Population Pharmacodynamic Models of Risperidone on PANSS Total Scores and Prolactin Levels in Schizophrenia"

_pharmaceuticals, 2024, doi:10.3390/ph17020148_

Round 1

Reviewer 1 Report

Comments and Suggestions for Authors

The authors should delineate the existing knowledge gap more explicitly in the Introduction, detailing what is currently unknown and has prompted this study.

Table 1 indicates the inclusion of schizophrenia patients with an average age of 18.3 years and a minimum disease duration of half a year. This may suggest a subset of early-onset cases, potentially adding to the sample's variability. Additionally, the table would benefit from an enhancement in quality, such as providing the interquartile range for median values.

The rationale for adopting a parameterized sigmoid Emax model in this study should be elaborated in the Introduction to aid reader comprehension. An explanation for selecting NONMEM 7 as the software of choice would also be informative.

The study currently considers only the aggregate PANSS total scores. However, the PANSS encompasses three distinct dimensions, each potentially exhibiting a varied response to treatment. An analysis disaggregating these dimensions would be insightful.

The identified correlation coefficient of 0.389 between the baseline score (E0) and the Hill coefficient (γ) indicates a moderate association. An elaboration on its clinical implications would enhance understanding.

The covariate analysis yields intriguing results. A discussion of the biological rationale and potential underlying mechanisms of these findings would be appreciated.

The paper would be enriched by proposing avenues for further investigation, especially those probing the underlying mechanisms of the observed phenomena.

The manuscript contains typographical errors, including the labeling in Figure 3, which need correction.

Comments on the Quality of English Language

Please proofread the text, making sure all the grammar and spelling are correct.

Author Response

Thanks for your comments. Responses was upload as an attachment.

Reviewer 2 Report

Comments and Suggestions for Authors

This is an interesting paper investigating the association of the prescription of risperidone, PANSS total scores and prolactin levels in schizophrenia.

The paper is well-written and of interest for the clinicians; however, several changes are recommended before considering it for publication.

ABSTRACT

1- I recommend to start with a brief description of the background to make context for the readers. 

2- Methods: please, about "by utilizing". What does it mean?

3- Results: How many patients were recruited?

INTRODUCTION

1- Please, introduce schizophrenia as a mental disorder with specific symptoms, phenotypes, etc. Not by the treatment used. I recommend to expand the introduction in terms of symptomatology.

2- Please, add more information about prolactin and its association with clinical outcomes.

RESULTS

1- A total of 144 patients were recruited. How many men, and women? Were women at the reproductive stage or postmenopausal? How many pre- and postmenopausal?

This is really important to discuss the findings.

2- Please, reinforme the idea of the relevance of using sigmoid E max models.

DISCUSSION

1-Limitations and strenghts should be highligjted and provided in a separate subsection (lines 186-194).

CONCLUSIONS

The conclusions section should not be a summary of the paper. Please, expand this section by adding some concluding remarks.

Author Response

Thanks for your comments. Responses were upload as an attachment.

Round 2

Reviewer 1 Report

Comments and Suggestions for Authors

My previous concern regarding the inclusion of early-onset schizophrenia patients in the study has not been addressed. Table 1 shows participants around 18 years old with a minimum disease duration of six months, indicating the presence of early-onset cases in the sample. This could contribute to increased variability. Prior research has highlighted notable differences between early-onset and adult-onset schizophrenia, such as in cognitive impairment and negative symptoms. This distinction should be considered in the analysis.

Concerning my previous inquiry about breaking down the three dimensions of the study, the scatter plot alone does not statistically validate that the observed improvements are predominantly due to positive scores and general psychopathology scores. Furthermore, the conclusion that "improvements in negative symptoms may require longer follow-up for observation" lacks clear rationale. Interpretations of the findings should incorporate existing literature on the efficacy of risperidone across different dimensions. Additionally, the significant decrease in PANSS positive scores might be influenced by the selection criteria of participants -- “patients were required to have a PANSS total score of 70 points, with at least two out of seven items on 100 the PANSS Positive Symptom subscale (PANSS-P) scoring 4 points.”. It seems like the authors picked those with severe positive symptoms, without similar emphasis on the other two dimensions.

The quality of figures and tables in the main text needs enhancement. Additionally, maintaining consistent formatting throughout would be beneficial.

It would be helpful if the authors could provide a summary of the treatment efficacy for this group prior to conducting population pharmacodynamic analyses. I am particularly interested in the PANSS reduction rate, typically calculated as (PANSS baseline score − PANSS end-point score)/(PANSS baseline score − 30) × 100%.

Comments on the Quality of English Language

It'd be great if the authors could make the manuscript more reader-friendly.

Author Response

Thank you for your comments. The point-by-point responses were uploaded as attachment.
